# Nucleophosmin: A Nucleolar Phosphoprotein Orchestrating Cellular Stress Responses

**DOI:** 10.3390/cells13151266

**Published:** 2024-07-27

**Authors:** Mohamed S. Taha, Mohammad Reza Ahmadian

**Affiliations:** 1Institute of Biochemistry and Molecular Biology II, Medical Faculty, Heinrich Heine University Düsseldorf, 40225 Düsseldorf, Germany; 2Research on Children with Special Needs Department, Institute of Medical Research and Clinical Studies, National Research Centre, Cairo 12622, Egypt

**Keywords:** apoptosis, DNA repair, molecular chaperone, NPM1, nucleocytoplasmic shuttling, nucleolar phosphoprotein, nucleophosmin, nucleoplasmin, stress response

## Abstract

Nucleophosmin (NPM1) is a key nucleolar protein released from the nucleolus in response to stress stimuli. NPM1 functions as a stress regulator with nucleic acid and protein chaperone activities, rapidly shuttling between the nucleus and cytoplasm. NPM1 is ubiquitously expressed in tissues and can be found in the nucleolus, nucleoplasm, cytoplasm, and extracellular environment. It plays a central role in various biological processes such as ribosome biogenesis, cell cycle regulation, cell proliferation, DNA damage repair, and apoptosis. In addition, it is highly expressed in cancer cells and solid tumors, and its mutation is a major cause of acute myeloid leukemia (AML). This review focuses on NPM1’s structural features, functional diversity, subcellular distribution, and role in stress modulation.

## 1. General Introduction

Nucleophosmin 1 (NPM1; also called B23 or Numartin) is one of the major nucleolar proteins, along with nucleolin and nucleostemin [1,2,3,4]. There are two other shorter isoforms, called NPM1.2 and NPM1.3 [5]. NPM1.3 lacks the C-terminal nucleolar localization signal (NoLS) and is mainly localized in the nucleoplasm [6,7,8,9,10,11,12]; its high expression is associated with many cancer types including lung adenocarcinoma [13]. NPM1 is the most studied and most abundant isoform [11]. NPM1 is a member of the nucleophosmin/nucleoplasmin family of nuclear chaperones, which includes nucleoplasmin 2 and 3 (NPM2 and NPM3) [14]. Since NPM1 has nucleic acid and protein chaperone activity, it is considered a molecular chaperone [15]. It interacts with almost all NoLS-containing proteins, such as Tat, CENP-W, p14ARF, and FMRP [11,16,17,18]. NPM1 shuttles between the nucleus and cytoplasm, and accordingly, a fraction of nucleolar NPM1 constantly translocates to the nucleoplasm and inner nuclear membrane, as well as to the cytoplasm and inner and outer plasma membranes [19,20,21]. Because of this ability, NPM1 has been implicated in many stages of viral infection through interaction with a variety of proteins from heterologous viruses [22,23], including human immunodeficiency virus type 1 Rev [20], human T-cell leukemia virus type 1 Rex [24], herpes simplex virus type 1 UL24 [25], bovine immunodeficiency virus chronic hepatitis B virus Rev [26], SAMD12-AS1 [27], circovirus PCV3 Cap [28], and chikungunya virus nsP3 [29].

NPM1 is a multifunctional protein involved in diverse biological processes, such as ribosome biogenesis [7,11,30,31,32], the maintenance of genomic stability [33], chromatin remodeling through histone chaperoning [34,35], centrosome duplication [36], and DNA repair [37]. NPM1 also plays a key role in response to various stress stimuli [38], including hypoxia [39], heat shock [40], oxidative stress [41], UV irradiation [42], chemotherapeutic agents [43], and gamma irradiation [44,45].

NPM1 is associated with various pathological conditions. It is frequently overexpressed, altered, rearranged, and sporadically deleted in human cancers [46]. Thus, there is little doubt that NPM1 is involved in human tumorigenesis, where its expression and gene integrity are frequently altered. NPM1 belongs to a novel gene category that functions as oncogenes and tumor suppressors. Depending on the expression level and gene dosage, either the partial functional loss of NPM1 or its aberrant overexpression can lead to neoplastic transformation through different mechanisms. Mutations in NPM1 occur in approximately one-third of patients with acute myeloid leukemia [47,48] and are clinically associated with leukocytosis, a high percentage of blasts, and extramedullary involvement [49]. In addition, NPM1 has shown increased interest in radiotherapy, where its knockdown significantly reduces tumor cell survival after irradiation. Irradiation has been shown to induce the dephosphorylation of NPM1 at T199, T234, and T237, and its intracellular distribution between the nucleoli, nucleoplasm, and cytoplasm [50]. Several studies have shown that the exposure of fibroblasts and lymphoblastoid cells to UV or γ-IR leads to the increased expression of NPM1 as an immediate early gene response, which is also induced by chromosomal instability [51,52]. Thus, NPM1 appears to be a key determinant of nuclear homeostasis in protecting cells from radiation-induced apoptosis, but the elucidation of the underlying molecular mechanisms awaits future investigation.

## 2. Structure and Functions of NPM1

NPM1 is the most abundant isoform with 294 amino acids (aa) and an approximate molecular weight of 37 kDa [22,47]. It consists of an N-terminal oligomerization domain (OD), a central histone binding domain (HBD), and a C-terminal nucleic acid binding domain (NBD) (Figure 1). 

The OD is highly conserved, methionine-rich, and contains a hydrophobic core [53] that organizes its oligomeric state. Thus, NPM1 exists in monomeric, pentameric, and decameric states [54]. The pentameric NPM1 is normally localized to the nucleolus and is involved in nucleosome formation and chromatin remodeling [11,55,56]. The N-terminal OD contains two canonical, leucine-rich nuclear export signals (NES; Figure 1) [16]. NES1 regulates ribosomal subunit binding, protein synthesis, and cell proliferation. NES2 binds the RAN complex with exportin 1 (XPO1; also called CRM1) and regulates NPM1′s association with centrosomes [16]. The mutation of a potential phosphorylation site at threonine 95 in NES2 abolishes the NPM1 association and inhibition of centrosome duplication [57]. The central unstructured, negatively charged HBD (Figure 1) mimics DNA structure for efficient binding to histones, preferentially to histone H3, to mediate nucleosome formation and chromatin decondensation [12,19,58,59]. The central HBD contains one of the two nuclear localization signals (NLS) that act as importin α/β recognition sites and mediate nuclear localization of NPM1. The C-terminal positively charged NBD preferentially binds RNA. It can also bind double-stranded DNA [60,61] and ATP [62]. In addition to an NLS, the NBD contains aromatic residues representing nucleolar localization signals (NoLS). Ribonucleolytic activity has been demonstrated at the 28S rRNA level in a common fraction of HBDs and NBDs [61,63], suggesting that NPM1 may play a role in ribosome maturation in addition to ribosome export from nucleoli.

Post-translational modifications, including phosphorylation, acetylation, ubiquitination, SUMOylation, and PARylation, modulate the subcellular compartmentalization of NPM1, thereby regulating its oligomeric state and functions (Figure 1) [31,54]. 

NPM1 is phosphorylated by several kinases, including casein kinase 2 [64], polo-like kinase 2 [65], cyclin-dependent kinase 1 [66], and the cyclin E-dependent kinase 2 (E/CDK2) complex [47,67]. Phosphorylated NPM1 at S48, S88, T95, and S125 is monomeric and nucleoplasmic [54,56]. The phosphorylation of T199 by E/CDK2 regulates mitosis, is responsible for centrosome duplication and pre-mRNA processing, and reduces the ability of NPM1 to bind RNA. T234 and T237 phosphorylation facilitates mitosis and detachment from the nucleolus. In contrast, the dephosphorylation of NPM1 by PP1β is involved in DNA repair and is induced by UV exposure [68]. Acetylation at lysines 54, 229, 230, 257, and 267 also modulates compartmentalized functions of NPM1, including interaction with RNA Pol II, and the regulation of DNA transcription by, for example, the Polycomb repressive complex 2 in the nucleoplasm [69,70,71,72]. Several processes, including HIV1 infection and drug resistance in cancer cells, depend on the induction and stabilization of NPM1 acetylation [73,74]. 

Ubiquitination and SUMOylation further control NPM1 localization and stability. BRCA1-BARD1 catalyzes the ubiquitination of NPM1 during mitosis, which appears to result in NPM1 stabilization rather than degradation [75]. BRCA1 forms a heterodimeric RING-type ubiquitin ligase with BARD1 to catalyze non-traditional K6-linked polyubiquitin chains [76]. BRCA1-associated protein 1 (BAP1) has been shown to disrupt BRCA1/BARD1 association and consequently inhibit NPM1 ubiquitination by BRCA1/BARD1 [77]. Mesencephalic astrocyte-derived neurotrophic factor (MANF) binds to NPM1. It increases the ubiquitination-mediated degradation of NPM1, leading to upregulation of the p53 signaling pathway and inhibition of cell proliferation, migration, and invasion ability in some cancer cells [78]. The SUMOylation of NPM1 is required for the recruitment of DNA repair proteins early in the DNA damage response, and SUMOylated NPM1 affects the assembly of the BRCA1 complex [79]. Sentrin/SUMO-specific peptidase 3 (SENP3) antagonizes p14ARF-mediated NPM1 SUMOylation to promote ribosomal biogenesis [79]. Following double-stranded DNA breaks, hCINAP is recruited to damage sites that promote the SENP3-dependent deSUMOylation of NPM1 [79]. In addition, p14ARF and TRIM28 cooperate to SUMOylate NPM1, thereby preventing centrosome amplification [80]. SUMOylation at K230 and K263 enhances Rb binding and E2F1-mediated transcriptional activity [81]. Porcine circovirus type 2 (PCV2) infection promotes the SUMOylation of NPM1 by activating the ERK/Ubc9/TRIM24 signaling pathway, resulting in the nucleolar regulation of PCV2 DNA replication [82]. NPM1 is associated with the poly(ADP-ribose) polymerase 1/2 (PARP1/2) [83]. PARP1 represses PD-L1 transcription through its interaction with NPM1 NBD, which is required for NPM1 binding to the PD-L1 promoter [84]. PARylated NPM1 has been reported [85], where NPM1 is known to be a target for mono- and poly-ADP ribosylation by several PARP proteins [29,66,86,87,88]. This PARylation modification regulates the myogenesis function for several proteins including NPM1 and human antigen R (HuR) [89]. The role of NPM1 PARYlation in regulating DNA damage repair foci [90] needs to be investigated in more detail.

## 3. NPM1 in Nucleolus

The nucleolus is a highly dynamic subnuclear compartment that undergoes major structural and compositional changes in response to growth signals, cellular status, and stress [3,4]. Various factors, including DNA damage, nutrient deprivation, viral infection, and exposure to certain chemicals, can induce nucleolar stress. The nucleolus is considered a stress-sensing organelle in cells, and genotoxic stress induces nucleolar dynamic and functional changes.

Specific cellular pathways aimed at maintaining cellular homeostasis and integrity are activated due to stress. Several diseases, including cancer and neurodegenerative disorders, have been implicated in the dysregulation of nucleolar stress responses [4,91]. The nucleolus is composed of three major compartments: the fibrillar center (FC; pre-rRNA transcription from rDNA), the dense fibrillar component (DFC; pre-rRNA processing), and the granular component (GC; ribosome unit assembly) (Figure 2) [92,93,94,95,96].

NPM1 is predominantly localized at the outer GC layer [97], where it regulates late rRNA processing and ribosome unit assembly [94,95]. Thus, the primary function of NPM1 is ribosome biogenesis [96,98,99]. In contrast, NPM1 directly binds and assists in the nuclear export of ribosomal protein L5 (rpL5) [100]. Therefore, blocking NPM1 affects the nuclear export of ribosomal protein L5 (rpL5) and 5S rRNA, reducing cell proliferation and cell cycle arrest. The endonuclease activity of NPM1 prevents it from cleaving and converting pre-rRNA to mature 28S rRNA [63,101,102]. NPM1 also shuttles and localizes premature 60S rRNA, 80S ribosomes, and polysomes [47].

NPM1 has been implicated in maintaining nucleolar structure by modulating liquid-liquid phase separation (LLPS) through mechanisms driven by electrostatic interactions between either the negative and positive domains on NPM1 (homotypic) or between the positively and negatively charged domain of NPM1 on one side with the positively charged R motifs on other nucleolar proteins or negatively charged rRNA (heterotypic) [11]. The interaction of NPM1 with SURF6 (Surfeit locus protein 6) is thought to modulate heterotypic vs. homotypic interactions [11,103,104,105]. Another nucleolar function of NPM1 is the initiation of the nucleolar stress response, where the nucleolus responds very early to stress and/or damage [2]; this early response occurs upon the modification of the nucleolar protein NPM1 by oxidation or S-glutathionylation (at cysteine residue 275), followed by the release and dissociation of the nucleolar protein NPM1 from the nucleolus and translocation to the nucleoplasm [48,106,107].

## 4. NPM1 in Nucleus

In the nucleus, NPM1 regulates cell survival through its interaction with PKB/AKT in response to growth factor stimulation and may begin to regulate and balance cell survival and apoptosis [108]. On the other hand, the inhibition of apoptosis is achieved by binding to both nuclear PI(3,4,5)P_3_ and nuclear AKT, a complex that directly interacts with caspase-activated DNase and inhibits its DNA fragmentation activity [32,33] This NPM1-dependent process appears to be controlled by nuclear PI3K and its upstream regulator PIKE (PI3K enhancer), which is a nuclear GTPase [32].

NPM1 contains two NES signals and its shuttling properties are regulated via its interaction with RAN-CRM1, which are mainly involved in the regulation of centrosome duplication and spindle assembly by cyclin-CDK complexes, p53, BRCA1, and BRCA2 (Figure 2) [46,57,109,110,111,112,113,114]. NPM1 acts as a regulator of cell cycle progression; its overexpression induces the rapid entry of hematopoietic stem cells into the cell cycle by inhibiting the expression of many negative cell cycle regulators during the G1-S phase, while NPM1 depletion upregulates the expression of these negative regulators and induces cell cycle arrest during stress [115]. The absence of NPM1 induces ploidy or uncontrolled centrosome amplification [114]. NPM1 is recruited to the centrosome through its interaction with BRCA2 and ROCK2 [116,117]. 

Interestingly, the phosphorylation of NPM1 is critical for regulating its nuclear functions. The phosphorylation of NPM1 on serine 4 by PLK1 and NEK2A regulates its reassociation with the centrosome during mitosis [118]. This process is also regulated by the NES of NPM1 and its interaction with RAN/CRM1, and the inhibition of its phosphorylation at T95, or mutation of the NES motif of NPM1, induces ploidy [6]. In addition, the phosphorylation of T 199 with E/CDK2 is required for centrosome duplication and the regulation of G2/M cell cycle arrest through its interaction with p53 and the regulation of its interaction with GADD45 [47]. Various stressors, such as UV, induce the rapid translocation of NPM1 to the nucleoplasm and its interaction with p53 and HDM2 [119]. NPM1 regulates the cell cycle by transporting hyperphosphorylated Rb to the nucleolus, allowing for the dissociation between Rb and E2F protein, where E2F allows the cell cycle to proceed into the S phase [120]. 

On the other hand, NPM1 regulates and maintains genomic stability by regulating many repair mechanisms of the DNA damage response. It also regulates centrosome replication [6,97]. NPM1 modulates base excision repair by forming a complex with the base excision repair proteins (APE1, FEN1, Polβ, and LIGL) [121,122], and the absence of NPM1 induces mislocalization between APE1, FEN1, and LIGL [12,97]. NPM1 affects the stability and localization of many base excision repair proteins, such as the alternative reading frame tumor suppressor p14ARF and the apurinic/apyrimidinic endonuclease 1 (APE1) [97]. Recently, it was shown that interaction between p14ARF and sirtuin7(SIRT7) blocks its interaction with NPM1 and subsequently induces its degradation via proteasomal activity [123]. Another way in which genome stability is regulated is through the regulation of DNA double-strand breaks by interacting with both p53 and its negative regulator, the E3 ubiquitin ligase HDM2. This interaction protects p53 from degradation and prolongs its half-life during stress [119]. NPM1 directly interacts with aa 639-1000 in BRCA2 and supports the double-strand break repair mechanism by forming BRCA2/RAD51 foci [116,124]. It interferes in translesion DNA synthesis by interacting with the catalytic domain of Pol ղ and preventing its proteasomal degradation [12,125]. The post-translational modification of NPM1 regulates its DNA repair functions, where phosphorylated NPM1 has been shown to translocate to DNA double-strand breaks with γ-H2AX foci [126], and the inhibition of this colocalization sensitizes cells to ionizing radiation [127]. 

Phosphorylated NPM1 recruits to double-strand break Rad51 foci, and its depletion does not affect the formation of the foci but leads to their persistence. The analysis of Rad51 and γ H2AX foci in NPM1-null mouse embryonic fibroblasts has been conducted, which are characterized by persistent DNA damage without affecting cell survival [126,128,129]. The dephosphorylation of NPM1 at threonines 199, 234, and 237 by the protein phosphatase PP1β due to the release of the retinoblastoma tumor suppressor (pRB) in response to UV irradiation is important for the activation of E2F1-dependent DNA repair mechanisms [48]. NPM1 SUMOylation at k263 by p14ARF is required for NPM1 involvement in HR and RAD51 foci formation [79,124]. 

## 5. NPM1 in Cytoplasm

Although NPM1 should shuttle between the nucleus and cytoplasm, its predominant presence in the cytoplasm indicates abnormalities or severe mutations, including cancer (acute myeloid leukemia, AML) and viral infection [24]. NPM1 in AML has insertional mutations that result in a frameshift that affects tryptophan residues on the nucleolar localization signal and converts it to a nuclear export signal, resulting in the aberrant cytoplasmic localization of NPM1 [16,48,130]. The latter can be inhibited by CRM1 inhibitors [131,132].

NPM1 may also mediate actin cytoskeletal dynamics through the Ras-dependent hyperactivation of the mammalian target of the rapamycin (mTOR) protein [133]. One of the most interesting functions of NPM1 is its role in regulating apoptosis. During stress, NPM1 suppresses the p53 apoptotic pathway by blocking the localization of p53 to mitochondria and preventing its translocation from the nucleus to mitochondria [37]. Another anti-apoptotic activity of NPM1 appears through reducing the proteolytic activity of several caspases (3, 6, 8), where a remarkable decrease in caspase 8 activity has been reported in cultured cardiomyocytes (Figure 2) [12,134]. On the other hand, NPM1 may regulate the pro-apoptotic activity in both intrinsic and extrinsic apoptosis. Its pro-apoptotic role in intrinsic apoptosis appears by its binding to activated BAX and the translocation of BAX to mitochondria. In this study, pro-apoptotic activity was monitored by detecting increased levels of mitochondrial cytochrome c release and caspase cascade activation with NPM1 downregulation using RNAi, and this pro-apoptotic effect is supported by the NFκB pathway and the direct interaction between NPM1 and other nucleolar proteins (RELA) [34,135,136]. Nevertheless, the oncogenic mutant of NPM1 impairs mitochondrial function [137]. Using pharmacological inhibitors of NPM1 to induce apoptosis in cancer cells, reports have monitored the anti-apoptotic properties of NPM1. The interaction of GAGE with NPM1 increases the stability of NPM1 and its resistance to interferon gamma-induced apoptosis [12,138]. In extrinsic apoptosis, NPM1 showed anti-apoptotic activity through its fusion with retinoic acid receptor alpha (RARA) and by blocking the TNF-mediated activation of caspases 3 and 8 (CASP3/8) through its interaction with the tumor necrosis factor receptor type 1-associated DEATH domain protein (TRADD) (Figure 2) [139,140].

Interestingly, the disruption of the oligomerization state of NPM1, by disrupting its pentameric state formation, induces apoptosis and affects its subcellular localization [134]. The inhibition of NPM1 nucleocytoplasmic shuttling also induces apoptosis [46,141,142]. All these data together suggest a bipartite role of NPM1 in apoptosis or as having an anti- or pro-apoptotic effect like some apoptotic marker proteins (e.g., BCL2) [143]. 

The N-terminal domain of NPM1 is structurally responsible for the binding of many viral proteins, such as HIV Tat and Rev proteins, herpes simplex virus US11 [20,22], hepatitis B core proteins [144], and adenovirus basic core proteins (Figure 2) [145]. As a result, NPM1 is involved in various stages of the viral life cycle, including the nuclear import of viral proteins and final assembly, making it a target for the treatment of various viral infections [11,23].

## 6. NPM1 at the Plasma Membrane and Its Secretion

The nucleolar proteins NPM1 and nucleolin have been reported at the inner leaflet of the plasma membrane using electron microscopy; these proteins have also been reported to interact with KRAS [146]. Another interesting point is that at the plasma membrane, NPM1 binds and stabilizes activated KRAS nanoclusters to modulate signaling through the MAPK pathway [35]. Furthermore, NPM1 regulates various cell signaling pathways such as proliferation and differentiation and cell survival via the MAPK pathway through its interaction with RAS proteins, specifically KRAS rather than HRAS. The acidic domain of NPM1 interacts with the basic domain of KRAS [35,146].

Several reports have drawn attention to the potential role of NPM1 in innate immunity. NPM1 was passively released into the extracellular milieu by necrotic or damaged cells but secreted by macrophages and monocytes [147]. This secretion can be accompanied by miRNAs, and NPM1 is responsible for the stabilization of the secreted miRNAs [148]. In addition, extracellular NPM1 acts as an inflammatory stimulator by inducing the production of inflammatory cytokines such as tumor necrosis factor (TNFα), IL-6, and IL-8 via ERK1/2 activation [149]. TLR4 is a receptor that mediates NPM1 signaling, which requires NPM1 binding to myeloid differentiation protein-2 (MD-2). Thus, NPM1 activity may be useful in the treatment of TLR4-related diseases [147].

NPM1 is considered one of the damage-associated molecular pattern (DAMPA) proteins, high mobility group box 1 (HMGB1); histones H3 and H4 are considered pro-inflammatory and cause cytotoxicity to living cells [149,150,151,152,153,154]. In addition, NPM1 can be secreted as a pro-inflammatory factor that induces migration and angiogenic regeneration by stimulating vascular endothelial growth factor-A (VEGF-A), hepatocyte growth factor (HGF), stromal-derived factor-1 (SDF-1), fibroblast growth factor-2 (FGF-2), platelet-derived growth factor-B (PDGF-B), and matrix metallopeptidase 9 (MMP9) properties in human endothelial cells [155], where it was reported that NPM1 interacts with TLR4 in these cells and activates an NFκB -dependent inflammatory pathway that upregulates interleukin IL-6 and COX-2 gene expression [156]. NPM1 regulates NFκB activity by binding its DNA-binding domain and enhancing its DNA-binding activity. NPM1 enhances inflammatory gene expression induced by tumor necrosis factor-alpha (TNF-α) and lipopolysaccharides (LPSs) in fibroblasts and macrophages [156].

## 7. Conclusions

NPM1 is an essential multifunctional nucleolar protein. It diffuses from the nucleolus during stress to interfere with many stress-related processes such as genomic stability, cell cycle, and apoptosis. Its aberrant cytoplasmic localization is a sign of cancer (acute myeloid leukemia). Recently, NPM1 has been identified as a key protein in either pro/anti-apoptotic activities and is responsible for the stabilization of many other proteins including tumor suppressor proteins; these findings developed our understanding of NPM1 beyond its implications in cancer prognosis and it has been identified as a primary therapeutic target. Approaches such as gene therapy to correct its mutations or targeting its interacting networks with small molecule inhibitors are being actively explored with a focus on cancer malignancy and cell response to radiotherapy. However, the molecular mechanisms underlying NPM1-related pathology are still poorly understood. Conversely, its anti-stress properties offer promising avenues for intervention in neurodegenerative diseases such as Huntington’s disease by managing pathological aggregates, as well as many other stress-related diseases.

Advances in the understanding of the mechanisms governing NPM1 secretion, its interaction with different extracellular receptors, its involvement in external apoptosis and its potential fusion with other proteins will open new avenues. These developments could potentially transform NPM1 from a therapeutic target into a bioengineered recombinant drug candidate for various diseases. In addition, considering its chaperoning properties and its ability to modulate different stages of viral cellular processing, all of this together suggests that NPM1 is a key regulator of innate immunity and may serve as an immunotherapeutic or pro-inflammatory agent in many diseases as well as viral infections.

## Figures and Tables

**Figure 1 cells-13-01266-f001:**
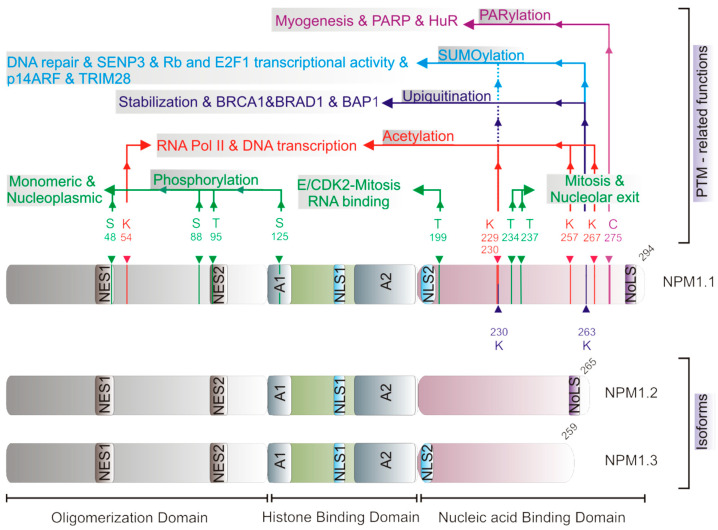
Functional domains, post-translational modifications, and associated functions of the NPM family. NPM1 is composed of three functional domains, the N-terminal oligomerization domain (OD), containing two nuclear export signals (NES1 and NES2), the central histone binding domain (HBD), comprising two acidic stretches A1 and A2 in addition to two nuclear localization signals (NLS1), and the C-terminal nucleic acid binding domain (NBD) containing the nuclear localization signal (NLS2) and the nucleolar localization signal (NoLS). Post-translational modifications (PTM) and related functions in addition to interacting partners, such as phosphorylation (green), acetylation (red), ubiquitination (dark blue), SUMOylation (light blue), and PARylation (magenta). For more details, see text.

**Figure 2 cells-13-01266-f002:**
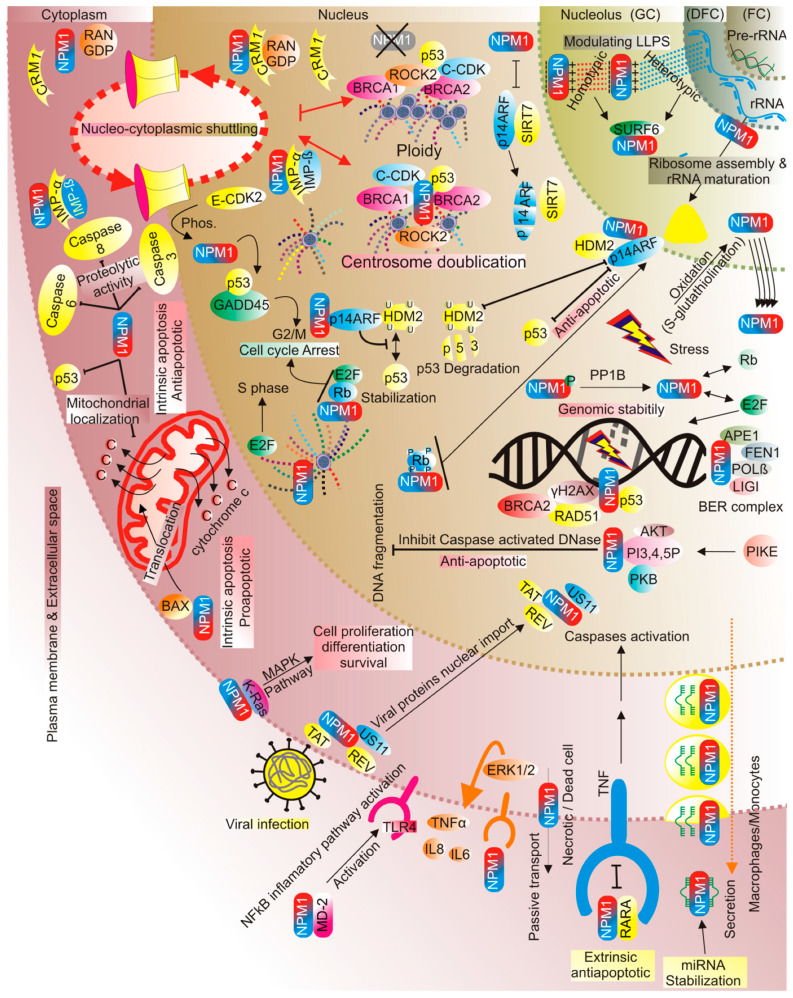
Schematic representation of the subcellular localization, associated functions, and interaction networks of NPM1. The nucleolus (fibrillar center (FC), dense fibril compartment (DFC), and granular compartment (GC)), nucleus(nucleoplasm), cytoplasm (cytosol), and outer layer represent the plasma membrane and extracellular milieu. For more details, see text.

## Data Availability

It is affirmed that no new data were generated while compiling this review manuscript. All referenced data sources are openly accessible and appropriately cited within the manuscript. Please do not hesitate to contact the corresponding author if any additional information or clarification is required.

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
