# Peer review of "Nucleophosmin: A Nucleolar Phosphoprotein Orchestrating Cellular Stress Responses"

_cells, 2024, doi:10.3390/cells13151266_

Round 1

Reviewer 1 Report

Comments and Suggestions for Authors

This is a well written overview of NPM1 function. 

This manuscript provides a general overview of the protein nucleophosmin (NPM1). Although there are a number of reviews on this topic, this manuscript is an excellent source of references (155 of them) that will direct researchers to delve deeply into NPM1’s biochemistry and cell biology. The manuscript begins with a general introduction that sets the stage for the remainder of the review. Next, the authors provide a section on the structure and functions of NPM1. This section details domains and PTMs that regulate function. Although the information is an overview, it is comprehensive in terms of describing PTMs. There is also an excellent figure that summarizes this information. The review proceeds to outline NPM1”s  subcellular localization and function in the nucleolus, the nucleoplasm and the cytoplasm. There is literature that indicates that NPM1 is secreted into the extracellular space and can function as a DAMPA/immune factor. The review provides a nice overview of this literature. A second figure illustrates the complexity of NPM1’s cellular roles. In summary, this review will serve as an excellent source of information for those new to the field of NPM1 biology.

Author Response

We sincerely thank the reviewer for taking the time and effort to read and review our manuscript. We are very pleased that the reviewer found the review to be of excellent quality.

Reviewer 2 Report

Comments and Suggestions for Authors

The review is well structured and sufficiently comprehensive, However it could be improved by better investigating the function of NPM1 as a vehicle of miRNAs in the extracellular environment.

Wang K, Zhang S, Weber J, Baxter D, Galas DJ. Export of microRNAs and microRNA-protective protein by mammalian cells. Nucleic Acids Res. 2010 Nov;38(20):7248-59. doi: 10.1093/nar/gkq601. Epub 2010 Jul 7. PMID: 20615901; PMCID: PMC2978372.

Author Response

We sincerely thank the reviewer for taking the time and effort to read and review our manuscript. Following the reviewer's suggestion, we have added the following text (line 296) on the function of NPM1 as a vehicle of miRNAs in the extracellular environment by also citing the study published by Wang et al. in Nucleic Acids Res. (2010).